# Tumor Volume Regression during and after Radiochemotherapy: A Macroscopic Description

**DOI:** 10.3390/jpm12040530

**Published:** 2022-03-26

**Authors:** Paolo Castorina, Gianluca Ferini, Emanuele Martorana, Stefano Forte

**Affiliations:** 1INFN, Sezione di Catania, 95123 Catania, Italy; paolo.castorina@ct.infn.it; 2Faculty of Mathematics and Physics, Charles University, V Holešovičkách 2, 18000 Prague, Czech Republic; 3REM Radioterapia SRL, 95029 Viagrande, Italy; gianluca.ferini@grupposamed.com; 4Istituto Oncologico del Mediterraneo, 95029 Viagrande, Italy; emanuele.martorana@grupposamed.com

**Keywords:** oncology, radiotherapy, biophysics, complex systems

## Abstract

Tumor volume regression during and after chemo and radio therapy is a useful information for clinical decisions. Indeed, a quantitative, patient oriented, description of the response to treatment can guide towards the modification of the scheduled doses or the evaluation of the best time for surgery. We propose a macroscopic algorithm which permits to follow quantitatively the time evolution of the tumor volume during and after radiochemotherapy. The method, initially validated with different cell-lines implanted in mice, is then successfully applied to the available data for partially responding and complete recovery patients.

## 1. Introduction

Lung and colorectal cancer are the first and second leading cause of cancer death, respectively [1]. In several studies, neoadjuvant radiochemotherapy has been shown to be both downgrading for nodal staging and beneficial in terms of overall survival [2,3,4]. Tumor regrowth is an important clinical parameters during chemo and radio therapy [5] and the probability of treatment benefit critically depends on the tumor progression pattern not only in the interval between the fractional doses but also at the end of the therapy [6]. The effect of preoperative radiochemotherapy in both lung and colorectal cancer improves survival and become the standard of care before tumor resection [7,8]. To evaluate the clinical results on the tumor volume reduction in radiotherapy (for constant cellular density systems), the tumor cell survival fraction, *S*, after *n* treatments at dose per fraction *d*, in the overall treatment time *t*, is usually written according to the linear quadratic model
(1)$$-ln\left(S\right)=n\left(\alpha d+\beta d^{2}\right)$$
where the tumor radiosensitivity is expressed by the parameters α and β.

Concerning the regrowth during radiotherapy, one has that:the untreated tumor growth has been usually described by means of the Gompertz law (GL) [9,10,11,12,13], a non linear growth pattern previously proposed in actuarial mathematics [14];in a transplantable rat tumor, it was shown that control and regrowth curves could be fitted by the same Gompertzian law [15];Gompertzian growth has been assumed to describe human tumor repopulation during fractional radiotherapy by Hansen et al. [16] and by O’Donougue [17].

Regarding chemotherapy, the GL regrowth requires a time increasing drug concentration to obtain an exponential depletion of the tumor cells: the Norton-Simon hypothesis [18,19]. In various tumor phenotypes the volume reduction still persists many weeks after the end of therapy and, for example, in colon-rectal cancer the best time for surgery has been estimated between 8–12 weeks after the completion of the neo-adjuvant radiochemotherapy. In other terms, at the end of treatment, tumor volume can increase or decline, according to specific microbiological conditions. Microbial community interacts with the host modifying its immune response, metabolism and oncogenesis [20]. Several studies on locally advanced rectal cancer (LARC) show that the persistence of a bacterium after neoadjuvant therapy is associated with a high relapse rates [21] and that alterations in the gut microbiome play an important role in predicting response to neoadjuvant therapy [22].

More generally, in in-vivo systems the time to tumor response to radiation therapy (RT) differs among different histologies for several factors depending on tumor location, oxygen landscape, vasculature, cell composition, etc. For example, tumors located in poorly lymph drained body sites (i.e., brain) or made of cells not migrating through lymph flow (i.e., some types of sarcoma) may develop a slower volume reduction following RT compared to tumors rich of lymph drainage (i.e., head and neck cancer, colorectal carcinoma, etc.), which acts as a scavenger of dead cells by means of immune system cells (i.e., lymphocytes, macrophages, dendritic cells, etc.) reacting against tumor cells damaged by radiation [23,24]. Indeed, the presence of dead material within a tumor could affect tumor dynamics in terms of volume loss following the administration of fractionated RT, as the tumor volume may even increase at the initial stages of treatment due to the coexistence of a proliferating cell proportion not yet adequately stopped by radiation and of an increasing dead material not yet adequately drained off [25,26]. Moreover, also the tumor vasculature is surely influenced by radiation, which can give rise to an ever-changing tumor oxygen landscape [27,28]. Endothelial cells swell as a consequence of radiation damage significantly reducing tumor oxygen supply. This unintentionally generates hypoxic cancer cells that are less prone to die when the subsequent radiation fractions are delivered. At these stages, on the other side, the removal of well-oxygenated cells re-enables the starved hypoxic cells to proliferation. At the end of treatment, it is expected that tumor vasculature is permanently disrupted so as to definitively impair tumor regrowth. This is another reason why tumor response assessment is several days after the RT finishes [29,30]. In clinical practice, when the classic fractionated RT is unable to remove all radioresistant niches, one assists to downsized still active tumors requiring salvage RT treatments [31,32]. Additionally, tumors are composed also of healthy cells surviving to RT. These could replace dead cancer cells or result in transforming fibrosis, which can alter tumor volume reduction and then perception of tumor response to RT. Lastly, the radiation-induced immune imprinting continues to work even several weeks after the end of RT treatment, being able to further downsize tumor until its disappearance when irradiation is ended in a long time [33,34]. Therefore the microscopic dynamics which produces the different evolution patterns of the tumor coarse-grain size during and after neo-adjuvant therapy is a complex phenomenon where different cell subpopulations (resistent and sensitive cells, apoptosis, necrotic core, …) are involved.

In this paper we propose a method, based on macroscopic variables [35,36] and with no explicit reference to the underlying dynamics, to analyze the quantitative evolution during and after the therapy. The starting point is the observation that the GL, initially applied to human mortality tables (i.e., aging), describes tumor growth. It has been also applied to kinetics of enzymatic reactions, oxygenation of hemoglobin, intensity of photosynthesis as a function of CO2 concentration, drug dose-response curve, dynamics of growth in bacteria and normal eukaryotic organisms and, more recently, in Covid-19 spreading. The GL, as other macroscopic growth laws, depends on two parameters, which for cancer are related to the initial exponential trend and to the maximum number of cell, $N_{\infty }$, called carrying capacity, that can be supported by the local microbiological conditions (angiogenesis, immune system, …). It is well known that the carrying capacity changes according to some “external” conditions in many biological, economical and social systems [37]. In tumor growth it is related to a multi-stage evolution [38]. In population dynamics, new technologies affect how resources are consumed, and since the carrying capacity depends on the availability of that resource, its value changes [38]. Therefore a simple method of monitoring the tumor evolution during and after radiochemotherapy is to understand how the carrying capacity (CC) changes, for the specific patient, due to the radiation and drug effects previously discussed [26]. This modification is difficult to predict, but different scenarios of regrowth (with different dependence of CC on the radiation dose, for example) are initially analyzed in the next sections both mathematically and by in vivo experimental data derived from mouse model, after tumor cell xenotransplantation, undergoing radiotherapy. Then the same algorithm is applied to the tumor volume regression pattern observed in rectal cancer during and after neoadjuvant radiochemotherapy for 23 patients [39,40]. The final results show how a modified carrying capacity is a useful macroscopic tool to describe the tumor size evolution during and after the end of therapy. The same conclusion is reached with the logistic growth law, showing the robustness of the proposed method. Moreover the approach is model independent, that is there is no assumption on the time evolution of the CC during and after therapy. On the other hand, more specific models consider the explicit time dependence of the CC by an additional differential equation, which requires assumptions and many more free parameters but have the advantage of a deep connection with the microscopic dynamics. In ref. [8], for example, the time evolution of the CC is directly related, through diffusion equations, with different aspect of vasculature (specific loss, stimulator capacity, inhibition, …) with a direct control on the effects of endostatin, angiostatin, and TNP-470 on tumor growth dynamics.

## 2. Materials and Methods

### 2.1. Macroscopic Growth Law and Carrying Capacity: General Formulas

The macroscopic growth laws for a population N(t) are solutions of a general differential equation that can be written as
(2)$$\frac{1}{N(t)}\frac{dN(t)}{dt}=f\left[N\left(t\right)\right]$$
where f(N) is the specific growth rate and its *N* dependence describes the feedback effects during the time evolution [35]. If f(N)= constant, the growth follows an exponential pattern. In particular, the Gompertz equations is
(3)$$\frac{1}{N(t)}\frac{dN(t)}{dt}=k\;\ln\frac{N_{\infty }}{N(t)}\;$$
where *k* is a constant with dimension (time${}^{-1}$) and $N_{\infty }$ is the CC, i.e., dN/dt=0 is reached when *N* is equal to $N_{\infty }$.

As discussed, the CC can be modified by effects not included in Equations (Equation 2) and (Equation 3). For example, the invention and diffusion of technologies lift the growth limit and the infectious diseases maximum spread increases by human mobility and by possible genetic mutations. Accordingly, let us assume that the dynamical effects which drive the time evolution of the population N(t) also modify the CC. During neo-adjuvant treatments, the condition, at t=0, for the GL evolution is given by the initial cell number, N(0). At time $t^{*}$, when the “external” perturbations end (i.e., the therapy), the initial number of cells, for the evolution for $t>t^{*}$, is $N(t^{*})$. Tumor regression depends on the sign of the specific rate (the second term in Equation (Equation 3)) which at the initial time can be positive (growth) or negative (depletion) if the ratio between the cell number and the CC, modified by the therapy with respect to its value without treatment, is <1 or >1. Therefore, the continuous regression during the treatment requires $N_{\infty }<N\left(t\right)$ for any *t* in the interval $\left[0,t^{*}\right]$, whereas after the end of therapy, for $t>t^{*}$, the condition is $N_{\infty }<N\left(t^{*}\right)$. In other terms, the ratio between the cell numbers and the modified CC has to satisfy the previous constraints.

### 2.2. Carrying Capacity and Radiotherapy

Let us first consider the CC change after the end of radiotherapy. Let us call $N(t^{*})$ the cell number after *n* doses, $N_{i}$ the initial cell number with $t^{*}=n\tau$, being τ the fractionization time. The subsequent evolution, after the *n* doses, starts with the initial condition $N(t^{*})$. Moreover, as previously discussed, also the CC changes due to some “indirect” effect. If $N_{\infty }$ is its value without treatment, the modified value, $N_{\infty }\left(d\right)$, depends on the dose *d*. In other words, the microbiological conditions of the environment have been modified and the evolution for $t>t^{*}$ depends on the ratio $r=N\left(t^{*}\right)/N_{\infty }\left(d\right)$. In particular if r>1 the tumor volume *V*, proportional to the cell number for homogeneous systems, decreases.

The solution of Equation (Equation 3) for the volume V≃N, for $t>t^{*}$ is given by (see Appendix A).
(4)$$\frac{V(t)}{V(t^{*})}=e^{ln\frac{V_{\infty }\left(d\right)}{V(t^{*})}\left[1-exp\left(-k\left(t-t^{*}\right)\right)\right]}$$

To avoid confusion, we define, $V_{\infty }^{NT}$, the volume carrying capacity without treatment.

### 2.3. Experimental Settings and Data Collection

The experimental process for the evaluation of in vivo lung cancer xenograft derived tumors in mice has been set as follows. Tumor material from human patients with primary Non-Small Cell Lung Cancer (NSCLC) have been obtained. The cancer cells was then isolated from tissue using mechanical and enzymatic digestion as described before [41]. Cancer cells where then cultured as 3d tumor-spheres using specific culture media. The four obtained cell lines were treated with different radiation gray (Gy) assessing proliferation, apoptosis and sphere formation capability in order to identify, for each line, the lowest dose of radiation capable of induce cell death in at least 50% of cultured cells (IC50). The four cell lines are then subcutaneously implanted into four groups of animals, each consisting of 19 immunodeficient female mice within 4 to 6 weeks of age. All mice injected with sphere forming cancer cells developed a subcutaneous mass between 1 to 3 months. In-vivo irradiation was performed using the lowest dose that impaired proliferation and increased apoptosis and the highest dose with a nonsignificant effect on proliferation and apoptosis. With this setting the doses administered were as follows: Line L1 with 5 and 8 Gy; line L2 and line L3 with 5 and 10 Gy; Line L4 with 8 and 10 Gy. The tumor volume was measured before the first irradiation session, after 15 days, and at the end of experiment, after 30 days. For each timepoint each mouse, in both control and study groups, was evaluated for tumor growth with an external caliper. Data are then stored in tabular form by line, radiation dose and mouse ID. To generate data useful for growth estimation, in this work we averaged all values for cell line and gray in order to get a general trend of tumor growth in mouse per line and radiation dose. The experimental results for untreated and treated tumor growth after different doses and for various cell lines are given in Table 1 for lung cancer, where the ratios $N\left(t\right)/N\left(t^{*}\right)$ is reported after 15 and 30 days from the end of the therapy.

### 2.4. Regression during and after Preoperative Radiochemotherapy for Rectal Cancer

Ref. [39] reports data on 15 patients with a biopsy-proven adenocarcinoma of the rectum who received standard preoperative radiochemotherapy treatment. Irradiation dose was 50 Gy in 25 fractions of 2 Gy, delivered in 5 weeks on workdays combined with oral capecitabine (XelodaVR, Roche, Basel, Swiss) b.i.d. (two times a day). 120 magnetic resonance imaging-(MRI) were acquired to follow the gross tumor volume (GTV) regression. The mean values of GTV are reported in Figure 1 and compared with the GL regression fit.

In ref. [40] data on tumor regression during and after radiochemotherapy from a prospective trial with 8 patients with locally advanced rectal adenocarcinoma have been analyzed. All patients received daily short magnetic resonance scans in treatment position prior to irradiation with the same immobilization equipment used for radiation therapy to image the morphological and functional changes of the tumorous tissue during the treatment. GTV was defined as the morphological tumor volume, visible on post-contrast computed tomography and co-registered MRI scans. All patients received 50.4 Gy in 28 fractions and concurrent chemotherapy with 5-fluoruracil (300 mg/m${}^{2}$ body surface area daily). Separate presentation of gross tumor volume dynamics for each patient during neoadjuvant radiochemotherapy for complete and partial pathological responders (pCR and pPR) are depicted in Figure 2 and Figure 3, compared with the GL behaviors (see Results and Discussion sections).

## 3. Results

The first step of our analysis is the evaluation of the GL parameters in NT cell lines in order to assess untreated tumor growth. Table 2 shows the parameters obtained for the volumes of untreated tumor after 15 and 30 days as they appear in Table 1. The GL parameters are fixed by data and the comparison of the phenomenological GL curves with data is depicted in Figure 4.

The observed moderate volume variations and the logarithmic dependence on the CC suggest to consider k≃ constant and, therefore, by a simple inversion of Equation (Equation 3), one derives the dependence of the volume carrying capacity, $V_{\infty }\left(d\right)$. The volume regression after the end of the irradiation treatment is, then, described by GL with a change in the volume CC. In Figure 5 the ratios $V_{\infty }\left(d\right)/V\left(t^{*}\right)$ as a function of *d* have been evaluated by data at 15 days and 30 days, showing complete consistency and the dose dependence of the modified carrying capacity. The responding (non-responding) cell lines have $V_{\infty }\left(d\right)/V_{\left(t^{*}\right)}<1$ ($V_{\infty }\left(d\right)/V_{\left(t^{*}\right)}>1$). Although few data points are analyzed, the result suggests to apply the proposed method to the larger in vivo data set in refs. [39,40].

The observed regression of rectal tumor during and after chemotherapy in ref. [39] can be directly fitted by GL with a volume CC less than the initial volume V(0). The comparison in Figure 1 gives a volume CC $V_{\infty }=0.292V\left(0\right)$, k=0.0285 in day${}^{-1}$.

The other data in ref. [40] report the volume reduction in rectal cancer during neoadjuvant radiochemotherapy for 8 patients, with partial (PR) and complete (CR) recovery. The time series cover about 28 days and the observed behaviour of each PR and CR patient is compared with the GL in Figure 2 and Figure 3. The corresponding GL parameters, given in Table 3 and Table 4, show a clear difference between PR and CR. Indeed for CR the CC is much smaller than the PR cases.

It should be recalled the Gompertz law parameters for many untreated tumor phenotypes are correlated [10,13]. More precisely, the two parameters, *k* and $ln(N_{\infty }/N_{0})$, should be anti-correlated. However this can be obtained only a-posteriori. A priori there is no clinical reason to expect the anti-correlation, which is still unexplained. From this point of view, the correct procedure is to consider a priori two different parameters since, moreover, the correlation has been observed for tumor growth without treatment or with radiotherapy, but our more relevant analysis concerns patients with neo-adjuvant radio and chemotherapy. Previous data can be analyzed by different macroscopic growth law. The logistic law (LL) (see Appendix A),
(5)$$V\left(t\right)=V_{\infty }\left[1+\left(\frac{V_{0}}{V_{\infty }}-1\right)*e^{\left(-\lambda t\right)}\right],$$
gives similar fits of the data, as reported in Figure 6 and Figure 7, with the values of the parameters given in Table 5.

## 4. Discussion

The application of coarse-grain algorithms to describe/predict the therapy effects on the time evolution of tumor size are often underestimated. The main reason is the difficult collection of experimental data in vivo, as evidenced by the few available data in literature.

On the other hand, our results pave the way for the construction of a, model independent, phenomenological approach to predict the shrinkage of the tumor size on the basis of a small number of data during the therapy or after its end.

For example, for the data in ref. [39], one can compare the observed data after the end of therapy with the prediction based on the CC fitted by data during radio and chemo treatment. By collecting data during the first 32 days, one predicts the evolution for almost two months later. The comparison is reported in Figure 8 and gives 0.33 (data) versus 0.23 (prediction) for the final observation time (day 89).

Analogously, for patient PR1, $V_{\infty }≃26$ cm${}^{3}$ and k=0.0825 per day, but one gets $V_{\infty }≃20$ cm${}^{3}$ and k≃0.07 per day by considering the first 11/23 data. In other terms, by a limited set of observations one could obtain a good approximation for the complete evolution.

This indication should be extremely useful for clinical decisions concerning modified treatments and/or surgery in a patient oriented way. However, a sufficient amount of data is required to disantangle the GL or the LL from an exponential or linear behavior and the evaluation of the smallest data subset to obtain reliable predictions is in progress.

## 5. Conclusions

Previous analyses suggest that the tumor regression volume during and after radiochemotherapy can be described by a simple and economical (2 parameters) way by the GL.

The macroscopic approach summarizes the complex underlying dynamics by the modification of the parameters, in particular the CC. Moreover the algorithm is patient oriented and, in this respect, it can be useful to have more quantitative hints on the best time for surgery in colon-rectal and other cancer phenotypes.

Finally, our findings are weakly dependent on the specific macroscopic law. Indeed we have repeated the analysis by the LL with similar conclusions, showing the robustness of the proposed method.

## Figures and Tables

**Figure 1 jpm-12-00530-f001:**
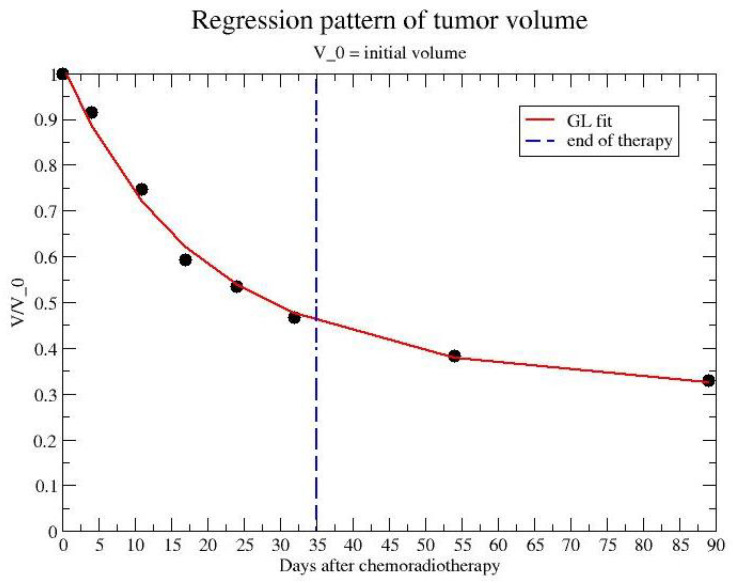
Comparison of the GL with modified CC with data on mean tumor volume regression during and after radiochemotherapy. dat from ref. [39]. Day 35 is the end of treatment. $V_{\infty }/V_{0}=0.292$, k=0.0285 in day${}^{-1}$, $\chi^{2}$ for d.o.f. =0.034.

**Figure 2 jpm-12-00530-f002:**
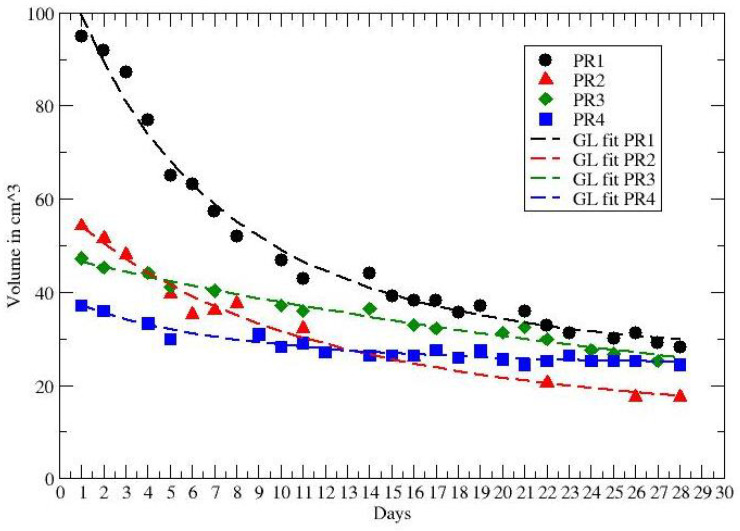
GL fit to the tumor volume regression for PR patients. Data from ref. [40].

**Figure 3 jpm-12-00530-f003:**
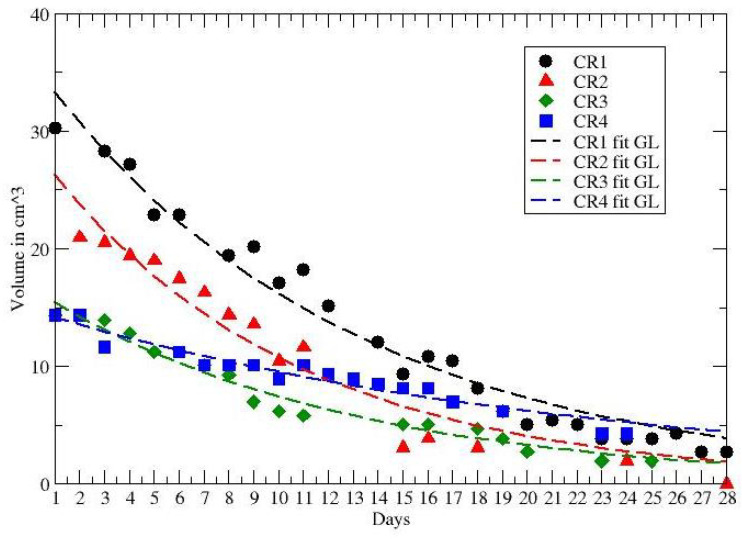
GL fit to the tumor volume regression for CR patients. Data from ref. [40].

**Figure 4 jpm-12-00530-f004:**
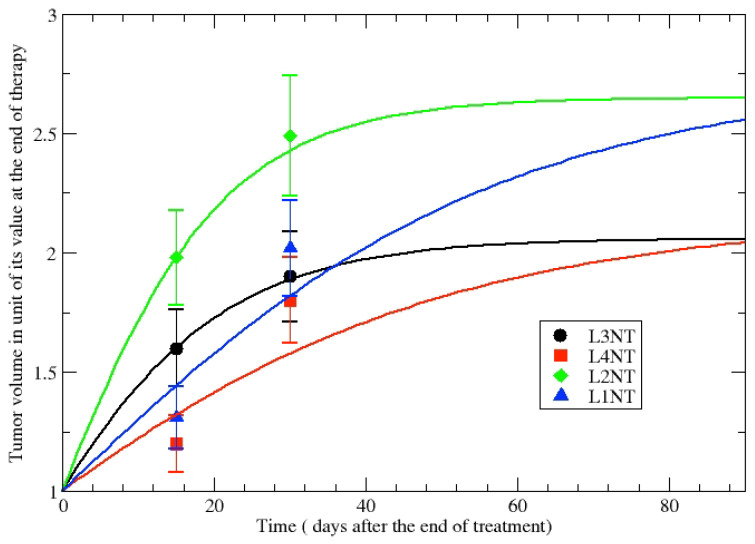
Comparison of the phenomenological GL curve with the untreated tumor volume evolution for different cell lines, see Table 2.

**Figure 5 jpm-12-00530-f005:**
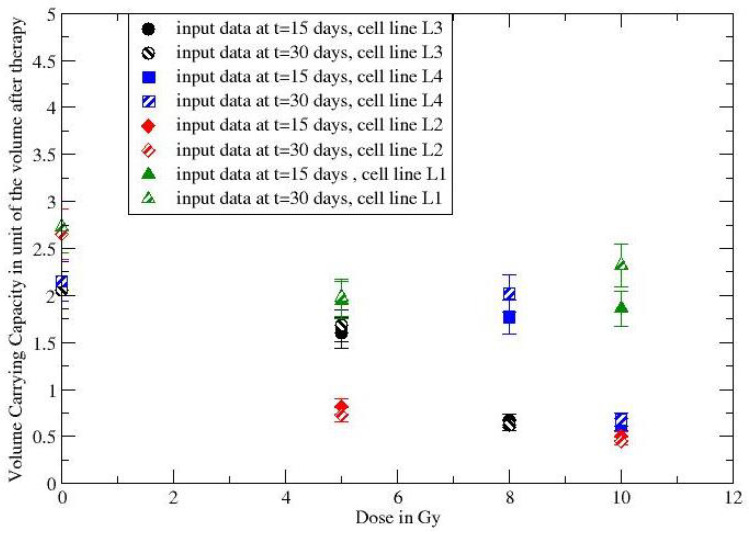
Dependence on the radiation dose of the volume carrying capacity for the experimental process with mice.

**Figure 6 jpm-12-00530-f006:**
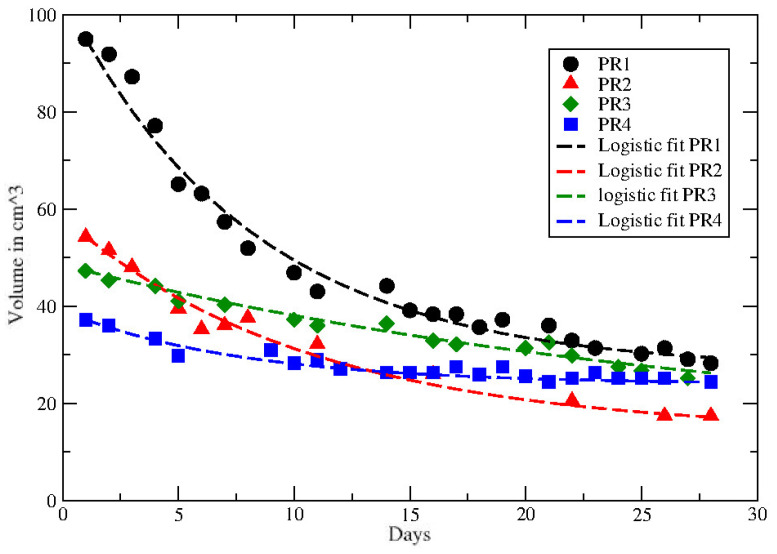
Logistic law fit to the tumor volume regression for PR patients. Data from ref. [40].

**Figure 7 jpm-12-00530-f007:**
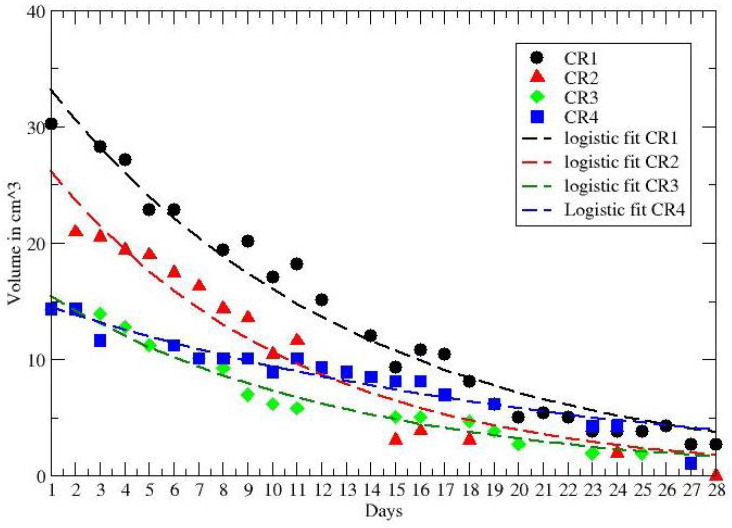
Logistic law fit to the tumor volume regression for CR patients. Data from ref. [40].

**Figure 8 jpm-12-00530-f008:**
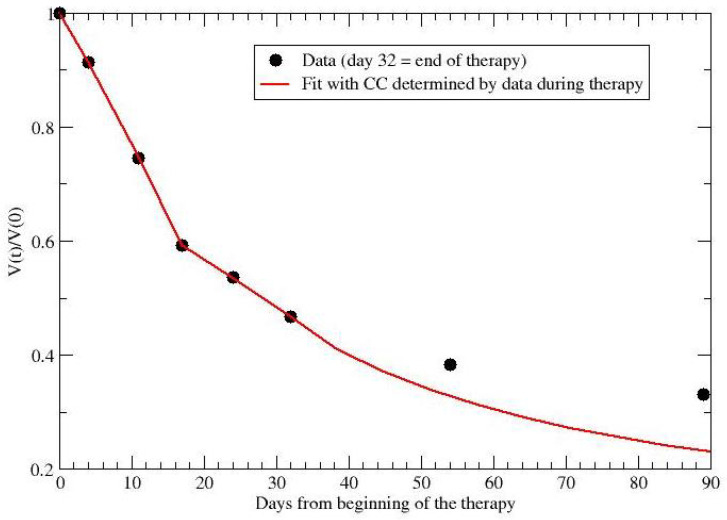
Comparison of data from ref. [39] with the GL parameters fitted during therapy.

**Table 1 jpm-12-00530-t001:** Lung cancer volume in unit of its volume at $t=t^{*}$ (i.e after radiotherapy). Without loss of generality, $V(t^{*})$ = 1 cm${}^{3}$ has been defined. An experimental error of about 10% has to be considered in the ratio corresponding to a precision measurement of 5%.

Cell Line/Dose	$t=t^{*}$ Day	$t=t^{*}$ + 15 Days	$t=t^{*}$ + 30 Days
L3-NT	1	1.6	1.9
L3-5Gy	1	1.36	1.58
L3-8Gy	1	0.77	0.66
L4-NT	1	1.2	1.81
L4-8Gy	1	1.23	1.79
L4-10Gy	1	0.85	0.8
L2-NT	1	1.98	2.49
L2-5Gy	1	0.87	0.75
L2-10Gy	1	0.62	0.49
L1-NT	1	1.32	2.02
L1-5Gy	1	1.27	1.51
L1-10Gy	1	1.25	1.65

**Table 2 jpm-12-00530-t002:** Carrying capacity and *k* parameter for untreated (NT) cell lines. An error of about 10% has to be considered. Fit of the cell lines L3, L2, L1 within 1 standard deviation (s.t.). Fit of the cell line L4 within 2 s.t., see Figure 4.

Cell Line	$V_{\infty }^{\mathrm{NT}}$ in cm${}^{3}$	*k* in Day${}^{-1}$
L3-NT	2.06	0.07
L4-NT	2.15	0.03
L2-NT	2.65	0.08
L1-NT	2.73	0.03

**Table 3 jpm-12-00530-t003:** GL parameters for CR patients. * = fitted value (see Appendix A).

Patient	$V_{\infty }/V_{0}$	*k* per Day	$k|\ln(V_{\infty }/V_{0})|$ per Day
CR1	<0.0896	<0.0335	0.0808 *
CR2	≃$10^{-7}$	≃0.006	0.1 *
CR3	<0.135	>0.041	0.0828 *
CR4	<0.075	>0.064	0.0426

**Table 4 jpm-12-00530-t004:** GL parameters for PR patients. * = fitted value (see Appendix A).

Patient	$V_{\infty }/V_{0}$	*k* per Day	$k|\ln(V_{\infty }/V_{0})|$ per Day
PR1	0.26 *	0.0825 *	0.11
PR2	0.21 *	0.0473 *	0.074
PR3	<0.53	>0.034	0.0218 *
PR4	0.64 *	0.107 *	0.048

**Table 5 jpm-12-00530-t005:** Logistic law parameters for PR and CR patients.

Patient	$V_{\infty }^{\mathrm{NT}}$ in cm${}^{3}$	λ per Day
PR1	27	0.123
PR2	14	0.094
PR3	12	0.0035
PR4	24	0.129
CR1	<5 × 10${}^{-7}$	0.0808
CR2	<5 × 10${}^{-7}$	0.1
CR3	<5 × 10${}^{-7}$	0.083
CR4	3.74 × 10${}^{-3}$	0.0483

## Data Availability

All data are available upon request, please contact the corresponding author.

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
