# Peer review of "Tumor Volume Regression during and after Radiochemotherapy: A Macroscopic Description"

_jpm, 2022, doi:10.3390/jpm12040530_

Round 1
Reviewer 1 Report
The paper is well written; authors should pay attention to the following points:
1- In table 1, write the unit of “t”.
2-In table 2, write the unit of “V” and “k”.
3-In table 5, write the unit of “V”.
4-In Figure 1, on the horizontal axis, “dose” should change to “Dose”.
5- In Figure 2, on the horizontal axis, “days” should change to “Days”.
6- In Figure 3, on the horizontal axis, “day” should change to “Days” and on the vertical axis, “volume” should change to “Volume”.
7- In Figure 4, on the horizontal axis, “day” should change to “Days” and on the vertical axis, “volume” should change to “Volume”.
8- In Figure 5, on the horizontal axis, “days” should change to “Days” and on the vertical axis, “volume” should change to “Volume”.
9- In Figure 6, on the horizontal axis, “days” should change to “Days”.

Author Response
1- In table 1, write the unit of “t”.
Table modified accordingly
2-In table 2, write the unit of “V” and “k”.
Table modified accordingly
3-In table 5, write the unit of “V”.
Table modified accordingly
4-In Figure 1, on the horizontal axis, “dose” should change to “Dose”.
Figure modified accordingly
5- In Figure 2, on the horizontal axis, “days” should change to “Days”.
Figure modified accordingly
6- In Figure 3, on the horizontal axis, “day” should change to “Days” and on the vertical axis, “volume” should change to “Volume”.
Figure modified accordingly
7- In Figure 4, on the horizontal axis, “day” should change to “Days” and on the vertical axis, “volume” should change to “Volume”.
Figure modified accordingly
8- In Figure 5, on the horizontal axis, “days” should change to “Days” and on the vertical axis, “volume” should change to “Volume”.
Figure modified accordingly
9- In Figure 6, on the horizontal axis, “days” should change to “Days”.
Figure modified accordingly
Reviewer 2 Report
This manuscript describes a mathematical modeling approach to simulate tumor volume regression during and after radiochemotherapy. This is a very timely subject and in line with the special issue of the journal on mathematical modeling of radiotherapy. Here, the authors focus on radiation reducing the carrying capacity in the Gompertz and logistic growth model – a concept introduce by Zahid et al. in a couple of papers during the past year – for murine data and a small data set of 15 rectal cancer patients.
While of high clinical importance, in its current form the manuscript has limited impact. Most importantly, too few details are given about the methodology to technically evaluate the approach, results, and conclusions. For example, it is unclear if each fraction of radiation causes a drop in carrying capacity, or only a single drop at the end that yields monotonic decline throughout therapy. Furthermore, there are large parts of the introduction that need references for the biological and clinical statements.
Mouse experiments appear to be a single dose of radiation, whereas clinical data appears to be fractionated radiation. It is unclear how acute radiation and fractionation are modeled differently.
Mouse data are averaged for cell lines and radiation doses. This means, the data and model derived parameters should be distributions, and then a form of Bayesian approach would be more suitable.
Table 2 should show units, and the fits should be shown together with an error quantification.
Parameter identifiability should be demonstrated. Following the cited work of Zahid, could this be reduced to a one-parameter problem? Currently, there appears to be overfitting.
Tables 3ff are confusing. Values are reported with 3-4 digits, but then as less than or greater than. Why are the actual values not reported?
In the discussion, the authors suggest that their work PREDICT THE SHRINKAGE of the tumor (line 188). This is incorrect. No prediction has been demonstrated, but mere fitting. The prediction study should really be done for clinical evaluation of the potential of the work.
Minor comments.
The seminal work of Hanfeldt et al. (Cancer Research, 1999) who introduced dynamic carrying capacities should be at least discussed.
The description of the Norton-Simon hypothesis appears incorrect. In brief, N-S states that treatment response should be proportional to the growth rate of the tumor. If the growth law is Gompertzian, treatment response should not be exponential (line 20).
Lines 65 to 68 appear to be unrelated to the manuscript and only serve as self-citations.
Line91. dN/dt=0 is reached when N=N_inf. That is mathematically incorrect. Even if N=inf., dN/dt>0.
103-105 The sentence is difficult to understand mathematically.
